

# On the role of trans-lithospheric faults in the long-term seismotectonic segmentation of active margins: a case study in the Andes

Gonzalo Yanez C.[1], Jose Piquer R.[2], Orlando Rivera H.[3]

[1] Pontificia Universidad Católica de Chile, Av. Vicuña Mackenna 4860, Macl, Santiago, Chile, gyaneza@ uc.cl

[2] Instituto de Ciencias de la Tierra, Universidad Austral de Chile, jose.piquer@uach.cl

[3] Minera Peñoles de Chile, orlando_rivera@penoles.com.mx

*Correspondence to*: Gonzalo Yanez C. (gyaneza@ uc.cl)

**Abstract.** Plate coupling play a fundamental role in the way in which seismic energy is released during the seismic cycle. This process includes quasi-instantaneous release during megathrust earthquakes and long-term creep. Both mechanisms can coexist in a given subducting margin, defining a seismotectonic segmentation in which seismically active segments are separated by zones in which ruptures stop, classified for simplicity as asperities and barrier, respectively. The spatiotemporal stability of this segmentation has been a matter of debate in the seismological community for decades. At this regard, we explore in this paper the potential role of the interaction between geological heterogeneities in the overriding plate and fluids released from the subducting slab towards the subduction channel. As a case study, we take the convergence between the Nazca and South American plates between 18º-40º S, given its relatively simple convergence style  and the availability of a high-quality instrumental and historical record. We postulate that trans-lithospheric faults striking at a high angle with respect to the trench behave as large fluid sinks that create the appropriate conditions for the development of barriers and promote the growth of highly coupled asperity domains in their  periphery. We tested this hypothesis against key short- and long-term observations in the study area, obtaining consistent results. If the spatial distribution of asperities is controlled by the geology of the overriding plate, seismic risk assessment could be established with better confidence.

## 1     Introduction

Subduction margins accommodate short-term (years to tens of years) and long-term (thousands to millions of years) deformation. The most evident effects of these two deformational behaviours are earthquakes (short-term) and mountain-building (long-term) (e.g. Avouac, 2007). The concept of the seismic cycle, introduced by Fedotov (1968) and further elaborated by Mogi (1977, 1985), identifies two stages: a long inter-seismic period (several tens of years), followed by a short co-seismic period (minutes at most) where the elastic energy stored



during the previous stage is released as an earthquake. For earthquake magnitudes in the range of Mw 7.5–9.5,
the observed mean slip displacement varies from 0.8–10 meters (Thingbaijam et al., 2017). Even though the
maximum mean slip in megathrust events is 10 meters, the zones of maximum slip, equated to asperities (e.g.,
Aki, 1984, Lay & Bileck 2007, Lay 2015) can reach 20–40 meters in wavelength patches in the range of 20–
100 kilometres (see, e.g., http://equake-rc.info/srcmod/). However, the release of elastic energy during the
seismic cycle only accounts for 90-95% of the deformation accumulated interseismically in convergent margins;
the remaining 5–10% produces permanent deformation in the overriding plate, expressed as crustal shortening
and mountain building (e.g. Yañez and Cembrano, 2004). This long-term process lasts for hundreds to
thousands of seismic cycles (time windows of millions of years). Therefore, both phenomena — earthquakes
and mountain building — are extreme responses to the same process: the convergence between oceanic and
continental plates, including the development of asperities and barriers in the same spatial and time frame.
The concepts of asperities and barriers were proposed by Lay et al. (1982) and Aki (1984) to describe the
process during the occurrence of an earthquake and intimately related to the concept of plate coupling More
recent studies (e.g. Bileck and Lay, 2007) propose a more complex mechanism at the subduction plate contact,
in which domains of unstable stick-slip state coexist with other domains in a conditionally stable stick-slip state,
and zones that develop aseismic slip/stable behaviour. These three states — unstable, conditionally stable, and
stable stick-slip behaviour — represent different slip modes that can be represented as asperities and barriers in
the old nomenclature. However, the conceptualization of Bileck and Lay (2007) proposes an along-dip (depth)
distribution of the different slip behaviours: (1) aseismic-stable at depths of 5–10 kilometres, (2) mostly
conditionally stable at depths of 10–15 kilometres, and (3) unstable stick-slip behaviour (Brace and Byerlee
(1966) and Burridge and Knopoff (1967)) at depths of 15–25 kilometres. Recent studies on exhumed subduction
domains in California (Platt et al., 2018) corroborate this along-dip transition from seismic zone to transition
zone. One interesting characteristic of these domains is that unstable domains are generally surrounded by
conditionally stable domains and aseismic domains in their outermost periphery.
To date, there is no clear evidence on whether the geological/tectonic process(es) control to some extent these
seismogeneic behaviours and/or their stability across several seismic cycles or geological time frames. Potential
candidates already proposed include: (1) the roughness of the subducting plate (aseismic ridges, fracture zones,
horst/graben structures, etc.) (e.g. Bilek et al., 2003, Wang and Bilek, 2011; Gersen et al., 2015; Philibosian and
Meltzner, 2020; Molina et al., 2021); (2) fluid-controlled overpressure (Peacock, 1990; Safer and Tobin, 2011;
Safer, 2017; Menant et al., 2019); (3) the shape of the subducting plate (e.g. Gutscher et al., 1999); (4) the
geology of the overriding plate (i.e Kimura et al., 2018; Philibosian and Meltzner, 2020; Molina et al., 2021),
among others, including various combinations of these different possible factors.
The role of fluids released from the subducting slab has emerged as a first-order factor in the plate-coupling
processes at subduction margins. Direct observations (e.g., Saffer and Tobin, 2011; Tsuji et al., 2014) and
numerical modelling (Menant et al., 2019) demonstrate that fluids released from the subducting oceanic crust
and subduction channel define segments at the plate-coupling zone with distinct pore pressure characteristics.
Overpressure domains are associated with zones of weak coupling, and strong coupling is observed in the case
of zones showing low pore pressure behaviour. The first type of domain is in direct association with creep zones



or slow slip events, while the other one is in direct association with locked zones, or in the seismological
nomenclature, the barrier and asperity domains, respectively. Seismic imaging of the forearc wedge (e.g. Tsuji
et al., 2014) and numerical modelling also show that fluids percolate upwards in the zones of maximum
overpressure, including the emplacement of serpentinite bodies along weak zones or faults.
In this paper, we propose a causal relationship between the presence of trans-lithospheric faults (TLF) in the
overriding plate and seismic segmentation, involving the control of TLF on the movement/storage/release of
overpressure fluids along and across the subduction zone. We use the Central Southern Andes as a case study,
as it is one of the most active seismogenic sites worldwide, is well studied, and has a relatively simple
subduction geometry (Hayes, 2018). In addition, recent structural and geophysical mapping has revealed the
role of TLF in the tectono-magmatic evolution of the continental margin of this region (e.g. Yanez et al., 1988,
Santibáñez et al., 2019; Cembrano and Lara, 2009; Melnick and Echtler, 2006; Yañez and Rivera, 2019; Piquer
at al., 2019, 2021a). We aim to demonstrate that the interaction between these TLF and the fluid circulating
through the subduction channel provides a simple first-order explanation for the Andean seismotectonic
organization through a long-lived geological control.
**2 Data and methods**
**2.1      Tectonic background**
The Nazca-South American plate convergence is a subduction-type margin that has been active in this segment
of the Andes since at least the Cretaceous without the accretion of new terrains (Mpodozis and Ramos, 1990).
Since 15 Ma, the convergence has been slightly oblique (E10°N) at a velocity of around 6.5 cm/yr (Angermann
et al., 1999). The age of the oceanic plate varies between 0 Ma at the triple junction of Taitao (44°S) to 45 Ma
at the Orocline bending of Bolivia (18°S) (Figure 1). A flat slab segment is located between 28°S and 33°S
latitude, affecting the development of an asthenospheric wedge landward and inhibiting the occurrence of active
volcanism since the last 5 Ma (Kay and Mpodozis, 2002). However, the Wadati-Benioff plane is roughly
homogenous in dip along the plate coupling between the Nazca and South American plates (Slab 2.0, Hayes,
2018). The roughness of the Nazca plate is affected by a progressively older oceanic crust northward, with some
fracture zones offsetting the plate, the subduction of a triple junction with an active spreading centre (now at
Taitao Peninsula), some episodic magmatic activity along the Juan Fernandez Ridge (33°S, Yáñez et al., 2001),
and eventually a smaller ridge at 20°S (Perdida Ridge, Cahill and Isacks, 1992). Overall, these features can be
described as minor obstacles to the subduction of a relatively young oceanic plate underneath a continental plate
in a highly coupled convergence margin (Section 2.5).
**2.2      Compilation of trans-lithospheric faults in the Andean active margin and their role as long-lived**
**high-permeability domains**





Trans-lithospheric faults (TLF) correspond to long-lived, high-angle fault systems, which have been identified
in several segments of the Andean margin, based on geological mapping (e.g. Santibañez et al. 2019; Cembrano
and Lara, 2009; Melnick and Echtler, 2006; Piquer et al., 2021a; Farrar et al., 2023; Wiemer et al., 2023), crustal
seismicity (e.g. Talwani, 2014,), a combination of indirect geophysical techniques (Yañez et al., 1998), or a
combination of all of these (Yañez and Rivera, 2019; Piquer et al., 2019; Pearce et al., 2020).
In Table 1 we present a synthesis of the current status of knowledge regarding TLF definition and the major
geological/geophysical evidences that described them. The number assigned in each case is used later on in
Figure 1 as an identificatory.
Detailed structural mapping in various segments of the Andean margin has provided direct geological evidence
for the presence of TLF. They are manifested in the field as networks of individual high-angle faults, defining
deformation zones with widths of up to several kilometres, and lengths in the order of hundreds of kilometres,
being possible to follow their trace across the entire continental margin (Lanza et al., 2013; Yáñez and Rivera,
2019; Piquer et al., 2021a). These fault networks correspond to the expression at the present-day surface of a
pre-existing TLF, as a result of its vertical propagation through Mesozoic and Cenozoic igneous and
sedimentary rocks (McCuaig and Hronsky, 2014; Piquer et al., 2019). Field observations also show that,
consistent with their high dip angle (commonly >60° and in several cases sub-vertical, although individual fault
segments can dip at slightly lower angles), TLF tend to be reactivated as basin-bounding faults during
extensional episodes, and are thus associated with violent changes in the stratigraphic record (Piquer et al.,
2015, 2021a; Yáñez and Rivera, 2019). They also control the distribution of exhumed basement blocks (Yáñez
and Rivera, 2019).
The geological record demonstrates that TLF are long-lived structures, which have played a major role in the
long-term evolution of the Chilean continental margin, being reactivated with different kinematics under
varying tectonic regimes. It is likely that several TLF were originated in the Proterozoic and the Palaeozoic
(Yáñez and Rivera, 2019); there is strong geological evidence suggesting the present-day TLF architecture was
already in place by the Permo-Triassic, a period in which these structures acted as master and transfer faults for
intra-continental rift systems (Niemeyer et al., 2004; Sagripanti et al., 2014; Espinoza et al., 2019). Syn-tectonic
emplacement of magma along TLF has been documented at least since the Jurassic (Creixell et al., 2011).
Geophysical support for the TLF architecture in the continental margin is provided by the geometry of magnetic
and gravimetric anomalies (Piquer et al., 2019; Yáñez and Rivera, 2019) and also by magnetotelluric data
(Pearce et al., 2020) and seismic tomography (Yáñez and Rivera, 2019). Evidence of seismic activity in some
of these TLF has been recorded, for example, a precursory event to the 9.3 Mw 1960 Valdivia Earthquake
(Lanalhue fault, Melnick et al., 2009), and the coseismic rebound associated with the 8.8 Mw 2010 Maule
earthquake (Pichilemu fault, e.g. Farías et al, 2011; Aron et al., 2013). Additionally, researchers have
documented a strong spatial relationship between a TLF and a major seismic swarm (Valparaíso seismic
sequence of 2017, Nealy et al., 2017) at the subduction megathrust (Piquer et al., 2021a).
Regarding the role of TLF as long-lived high-permeability domains, Yañez and Rivera (2019) postulated that
they represent weak lithospheric domains that favour fluid flow and the emplacement of different types of ore
deposits over large time periods (tens of millions of years), beginning with stratabound and IOCG-type deposits





in the Jurassic. A similar conclusion has been reached by Farrar et al. (2023) for the emplacement of porphyry
copper deposits of various ages, and by Wiemer et al. (2023) for gold-rich superclusters of various types of
mineral deposits. The strong relationship between the locations of TLF and those of giant ore deposits at specific
metallogenic belts has been discussed more specifically in the Andes of Northern (e.g., Chernicoff et al., 2002)
and Central Chile (e.g., Piquer et al., 2016) and neighbouring regions in Argentina. Similarly, there is a well-
established relationship between the locations of TLF and volcanic/geothermal activity in the Andes of Southern
Chile (e.g., Cembrano and Lara, 2009). Moreover, high $Vp/Vs$ ratios that were documented during the
Pichilemu seismic sequence following the 2010 Maule earthquake have been interpreted as strong evidence of
fluid migration (Farías et al., 2011).
Various authors have discussed how the type of magmatic-hydrothermal product and fluid flow regime varies
depending on the orientation of a specific high-angle fault system (in several cases, a TLF) relative to the
predominant stress tensor (Lara et al., 2006; Cembrano and Lara, 2009; Roquer et al., 2017; Piquer et al.,
2021b). Of particular relevance is the orientation of the fault system relative to the maximum stress ($\sigma 1$); if the
fault system is sub-parallel or strikes at a low angle relative to $\sigma 1$, it is well-oriented for opening and reactivation
respectively, allowing the rapid ascent of magma and hydrothermal fluids through different crustal segments.
On the other hand, if the fault system is sub-perpendicular or strikes at a high angle relative to $\sigma 1$, it would be
poorly oriented or misoriented for reactivation and would promote the storage of magma and hydrothermal
fluids at depth (e.g. Cembrano and Lara, 2009; Stanton-Yonge et al., 2016; Piquer et al., 2021b). In the latter
case, a requirement for fault reactivation and the release of the accumulated fluids is that supra-lithostatic fluid
pressures are achieved; once this occurs, the fault system would allow the discharge of the accumulated fluids
towards upper crustal levels and would act as a fluid pump ("fault-valve behaviour"), concentrating fluids in
the fractured areas within the fault system and leading to the depletion of fluids in the surrounding regions
(Sibson, 1990, 2020; Cox, 2016). These fluid discharge events cause seismic swarms (Cox, 2016), which
concentrate at the base of the high-angle fault system (Sibson, 2020).
Figure 1 presents the main array of NW- and NE-striking TLF observed in the Andean margin; their seaward
trend has been extrapolated following the observed trend in the continental lithosphere, in particular south of
36□S, following the trace of submarine canyons.



Table 1: Main Trans-Lithospheric Faults of the Chilean Andes (17-42°S Latitude)

| LSS_ID | LSS_NAME | REFERENCES | GEOLOGICAL EVIDENCES |
|---|---|---|---|
| 1 | Visviri | (15), (22) | (L)(TLS)(SC)(GVA), Antofalla Basement (T) |
| 2 | Arica | (15), (21), (22) | Arequipa Massif (T), ETL NW Arica (TLS) |
| 3 | Camarones | (22) | (TLS)(GVA)(SDGU) |
| 4 | Iquique | (22) | (TLS)(GVA)(SDGU) |
| 5 | Calama | (1), (2), (8), (20), (21), (22), (24) | Comache (F), Calama-Olacapato-El Toro (L)(FS)(VA), Solá (F), Chorrillos (F), ETL NW Calama (TLS) |
| 6 | Mejillones-Llullaillaco | (8), (10), (21), (22), (29) | Archibarca (L)(VA), Cataclasitas de Sierra de Varas (DZ), ETL NW Mejillones (TLS) |
| 7 | Agua Verde-Exploradora | (8), (22) | Culampajá (L) |
| 8 | Antofagasta-Conchi | (22), (27), (28), (30) | Antofagasta-Calama (L)(PMG)(TR)(STMH) |
| 9 | Taltal-Potrerillos | (8), (22) | Taltal (L)(TLS)(SDGU)(VA)(STMH) |
| 10 | Chañaral | (8), (22) | (TLS)(GVA)(SDGU) |
| 11 | Copiapó | (22) | (TLS)(SDGU) |
| 12 | Vallenar | (22) | (TLS)(SDGU) |
| 13 | Domeyko | (22) | (TLS)(GVA)(SDGU) |
| 14 | Vicuña | (22) | (TLS)(GVA)(SDGU) |
| 15 | Andacollo | (22) | (TLS)(GVA)(SDGU)(STMH) |
| 16 | Punitaqui-Los Pelambres | (22) | (TLS)(GVA)(SDGU)(MA)(STMH) |
| 17 | El Potro | (22) | (TLS)(SDGU) |
| 18 | Illapel | (22) | (TLS)(GVA)(SDGU) |
| 19 | Almendrillo | (22) | (TLS)(GVA)(SDGU) |
| 20 | La Ligua-Los Andes | (21), (22), (31) | (TLS)(GVA)(SDGU)(SC)(MA), Río Blanco-Los Bronces (FS)(STMH) |
| 21 | Valparaíso-Volcán Maipo | (3), (5), (7), (19), (21), (22), (23), (26) | Piuquencillo (F)(FS)(STMH), Melipilla (F)(MA), Marga-Marga (FS), Valparaíso-Curacaví (FS)(STMH), Concón (MDS), Cartagena (MDS), El Tabo (MDS) |
| 22 | Pichilemu | (9), (17), (22), (23), (24), (25) | Pichilemu (ATS), Teno (FS)(SC)(STMH), Planchón-Peteroa (LLBS)(SC) |
| 23 | Laguna del Maule | (32), (33) | Río Maule (F)(VA)(SDGU)(STMH) |
| 24 | Iloca-Rio Melado | (34) | Laguna Fea (FS)(VA)(STMH) |
| 25 | Aconcagua-San Antonio | (4), (6), (22), (23), (31) | Puangue (F), Estero Chacabuco (F), Estero Colina (F), El Salto (FS)(STMH) |
| 26 | Volcán Quizapu | (33) | (VA)(MDS) |
| 27 | Parral-Bullileo | This study | (VA)(SDGU) |
| 28 | San Carlos-Nevados de Chillán | (12), (17), (18) | Chillán (AZ), Nevados de Chillán-Tromen (LLBS), Cortaderas (L) |
| 29 | Lanalhue-Volcán Villarrica | (11), (14), (16), (17), (24) | Morguilla (FLS), Lanalhue (F)(FS), Villarrica-Quetrupillán-Lanín (LLBS) |
| 30 | Tirúa-Pitrufquén | (11), (16) | Mocha-Villarrica (FS) |
| 31 | Rio Calle Calle-Lago Ranco | (13), (17) | Carrán-Los Venados (LLBS), Futrono (F) |
| 32 | Puerto Saavedra-Volcán Callaqui | (18) | Copahue-Callaqui (AZ) |
| 33 | Osorno-Volcán Calbuco | This study | (VA) |
| 34 | Ancud-Volcán Michimahuida | (17) | Michimahuida (LLBS) |
| 35 | Cucao-Chaitén | (17) | Chaitén (LLBS) |
| 36 | Chacao-Osorno-Puntiagudo | (17) | (VA) |

Abbreviations: (ATS) Andean Transverse System; (AZ) Accommodation Zone; (DZ) Deformation Zone; (F) Fault; (FLS) Fault-line Scarp; (FS) Fault System; (GVA) Gravimetric Anomaly; (L) Lineament; (LLBS) Long-Lived Basement Structures; (LLTF) Long-Lived Transverse Fault; (MA) Magnetic Anomaly; (MDS) Mafic Dike Swarm; (PMG) Paleomagnetism; (SC) Seismic Cluster; (SDGU) Structural Discontinuity of Geological Units; (T) Terrane; (TLS) Translithospheric Structures; (TR) Tectonic Rotations; (STMH) Syn-Tectonic Magmatic-Hydrothermal Centers; (VA) Volcano Alignment. Reference Keys: (1) Salfity, 1985; (2) Marrett et al., 1994; (3) Gana et al., 1996; (4) Wall et al., 1996; (5) Yáñez et al., 1998; (6) Wall et al., 1999; (7) Rivera & Cembrano, 2000; (8) Chernicoff et al., 2002; (9) Sernageomin, 2003; (10) Niemeyer et al., 2004; (11) Haberland et al., 2006; (12) Ramos & Kay, 2006; (13) Lara et al., 2006; (14) Glodny et al., 2008; (15) Ramos, 2008; (16) Melnick et al., 2009; (17) Cembrano & Lara, 2009; (18) Radic, 2010; (19) Creixell et al., 2011; (20) Lanza et al., 2013; (21) Rivera, 2017; (22) Yáñez & Rivera, 2019; (23) Piquer et al., 2019; (24) Santibáñez et al., 2019; (25) Pearce et al., 2020; (26) Piquer et al., 2021a; (27) Arriagada et al., 2003; (28) Peña, 2010; (29) Richards et al., 2013; (30) Palacios et al., 2007; (31) Piquer et al., 2015; (32) Kohler, 2016; (33) Fischer, 2021; (34) Torres, 2021.




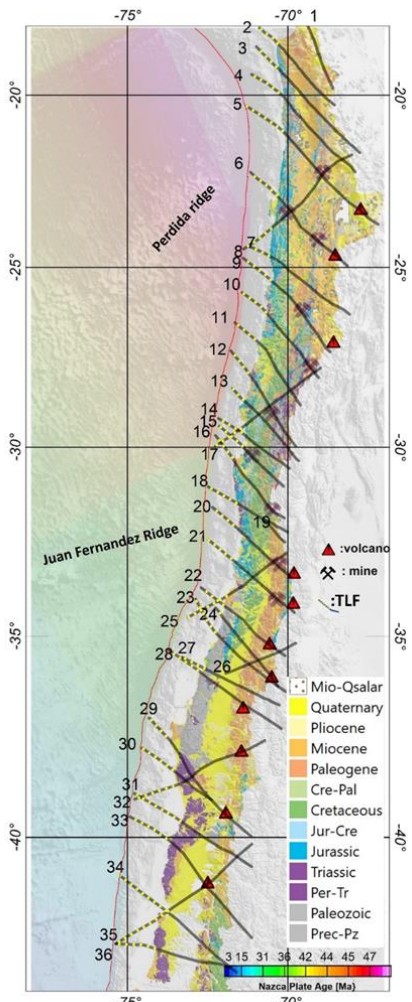


**Figure 1: The spatial distribution of trans-lithospheric faults (TLF) over the regional geology of the Chilean continental margin (from SERNAGEOMIN, 2003). The traces of the TLF's are based on the models of Yáñez and Rivera (2019) and Piquer et al. (2019) in Northern and Central Chile, and after the model of Melnick and Echtler (2006) in Southern Chile. Also shown are the locations of the main ore deposits (from north to south, Chuquicamata, Mantos Blancos, Escondida, Salvador, Cerro Casale, El Indio, Andacollo, Los Pelambres, Río Blanco-Los Bronces and El Teniente), and active volcanoes (from north to south, Láscar, Llullaillaco, Ojos del Salado, Tupungatito, Maipo, Planchón-Peteroa, Laguna del Maule, Chillán, Callaqui, Villarrica and Osorno) to show their correspondence with the TLF array. TLF are extended until the trench, following their main trend and the canyons trace to the south of 36□S, using segmented red lines to highlight the uncertainty of this offshore extension. In the seaward side of the figure, the age map of Müller et al. (2019) is included with the bathymetry of the seafloor**





**2.3    Historic seismicity and Slip solutions during the last 50 years Trans-lithospheric faults (TLF) correspond**

The historic seismic record in the region is short, extending from the start of the Spanish Colonization in the region (present territories of Perú and Chile, circa 1500 ac). Compilations of historic seismicity and subsequent interpretation to assess the magnitude and longitudinal extension of the events have been provided in Ruiz and Madariaga (2018) and Scholz and Campos (2012), among others. Figure 2, panel A, includes all the historic events described by these authors, as well as events above 7 Mw from the USGS catalogue. As noted by several authors (Ruiz and Madariaga, 2018, and references therein), there is evidence of seismo-tectonic segmentation in the historic record. For the present analysis, we define seven domains from north to south with some distinctive characteristics; the boundaries between each domain are defined by a width domain of roughly 100-200 kilometres that represents the uncertainty in the rupture length of the major events,  wider boundaries for the cases of lacking information, in particular in the northern area where the historic record is scarce  Domain I, in the northernmost part of the study region, shows a sequence of events close to magnitude 8 Mw and separated by 100–150 years. Domain II has no large events (above 8 Mw) in the historic record, instead having a series of intermediate events of magnitude 7–7.7 Mw between 1960 and 2020. Domain III has two events with magnitudes in the range 8.3–8.5 Mw separated by almost 10 years, but with a current seismic gap of 100 years. Domain IV is less than 200 kilometres in length and includes a series of seismic events of magnitude 8 Mw or above. According to Ruiz and Madariaga (2018), the three major events in this domain show relatively consistent recurrence times (60–80 years) and magnitudes (8–8.4 Mw), namely, the earthquakes of 2015 (Illapel, 8.3 Mw), 1943, and 1880. Domain V is also relatively small, about 300 km, and includes regular events of around magnitude 8 Mw, including the Valparaiso 1985 8 Mw event and the 1906 8.4 Mw event.  Domain Vi, VII and VIII include part of the Maule 2010 8.8 Mw and Concepción 1835 8.6 Mw events, but are defined as such based on some less than 8 Mw events, Domain X, the southernmost domain, is dominated by the giant events of Valdivia 1960, 9.3 Mw, and 1737, 9.0 Mw.

Adequate seismic coverage is available since 1985 in Chile. In this period, six large earthquakes have been recorded: Valparaiso 1985, 8.0 Mw (Comte et al., 1986; Mendoza et al., 1994); Antofagasta 1995, 8.0 Mw (Ruegg et al.; 1996, Delouis et al., 1997; Pritchard et al., 2002 and Chlieh et al., 2004); Tocopilla 2007, 7.8 Mw (Schurr et al., 2012); Maule 2010, 8.8 Mw (Delouis et al., 2010; Lay et al., 2010; Vigny et al., 2011; Koper et al., 2012; Ruiz et al., 2012; Moreno et al., 2012; Lorito et al., 2011; Lin et al., 2013; Yue et al., 2014); Iquique 2014, 8.2 Mw (Ruiz et al., 2014; Hayes et al., 2014; Schurr et al., 2014; Lay et al., 2014), and Illapel 2015, 8.3 Mw (Melgar et al., 2016; Heidarzadeh et al., 2016; Li et al.,2016; Lee et al., 2016; Satake and Heidarzadeh, 2017). Given the large size of the Valdivia 1960 earthquake (9.3 Mw), we also include slip estimates for this event based on surface deformation data (Barrientos and Ward, 1990). The slip distribution of these events ranges from 1 meter (e.g. Tocopilla 2007, Antofagasta 1995), several meters (e.g. Illapel 2015, Iquique 2014), and more than 10 meters (Valdivia 1960, Maule 2010); however, in Figure 2, panel B, we normalize the slip



distribution with respect to the corresponding maximum slip in each case, plotting over the slab surface to
highlight its spatial distribution. This approach aims to highlight the zones of maximum slip in each case and
to appreciate their spatial and temporal distribution, under the working hypothesis that they represent the zones
of maximum slip and are most likely a good proxy to identify asperities in the plate contact zone. These
maximum slip zones are generally distributed between the TLF network (Figure 2).

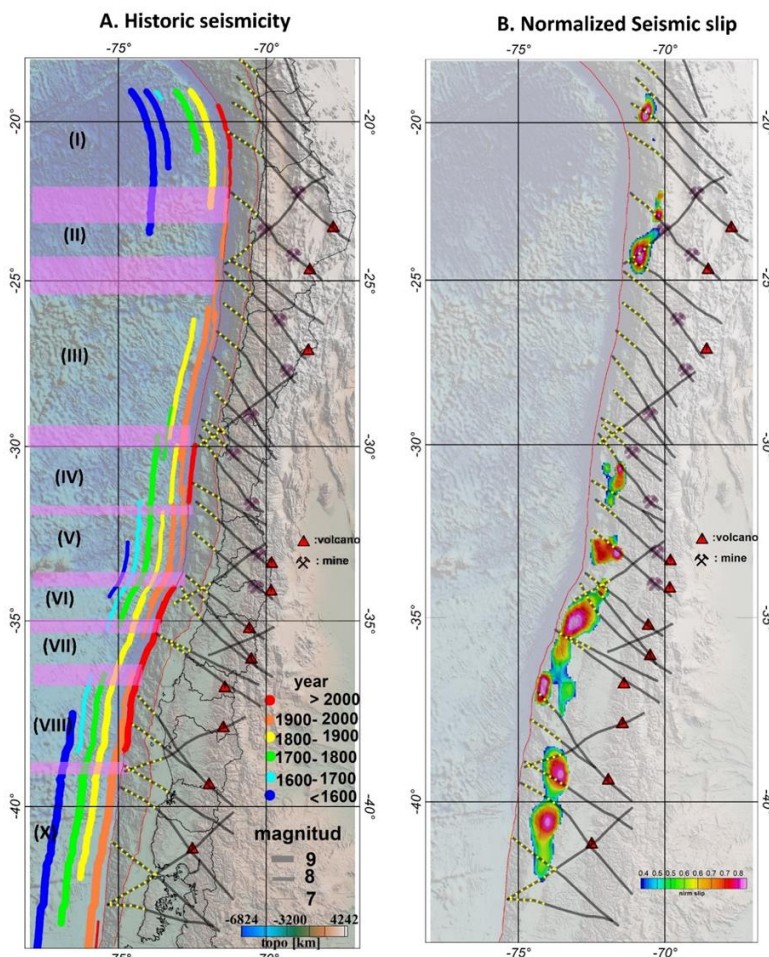


**Figure 2: Panel A: historical seismicity from the years 1450 to 2020 (for a full description of each event, see Table
A.1 of the supplementary material). The lateral extent of each event indicates the NS estimate of the event name; the
colour scale corresponds to the Mw magnitude in each case. Seismo-tectonic segmentation is indicated by yellow
semi-transparent ribbons, which are extended downwards to the lower panels. Panel B: zones of maximum slip in
the megathrust events registered at the margin of Chile since 1960.**

**2.4    Cumulative seismic spatial distribution in the last 20 years**

The seismic activity, apart from its spatiotemporal distribution around megathrust events (occasionally with
foreshocks and normally with a hyperbolic distribution of aftershocks in time (Omori's law)), shows some



clustering (denominated in general as seismic swarms), that may be triggered by aseismic creep events
(Forsyth et al., 2003; Roland and McGuire, 2009) associated with the presence of fluids in the fault zone. In
the Andean plate convergence margin, recent studies also show examples of seismic swarm distribution
attributable to fluid pore-pressure processes (e.g. Poli et al., 2017, Pasten-Araya et al., 2018). To contextualise
the spatial distribution of this seismicity, we compute a normalised seismic density distribution along the
margin for the last 20 years in which the seismic network is complete above magnitude 3 Mw. We exclude
most of the seismicity associated with major thrust events in this period, filtering out the events at distances of
less than 200 kilometres from the rupture zone in a temporal window of 200 days. We acknowledge that this
20-year time window is too short to obtain a broad and complete picture of the distribution of swarms along
the margin. However, as swarms normally last for just a few weeks or 1–2 months at most, the cases observed
in this time window provide insights into their spatial distribution. The data used in this analysis were
obtained from the database of the National Seismological Centre (CNS in Spanish). We selected data
attributable to the seismogenic plate contact within a 10-kilometre-thick volume following the slab 2.0
Wadati-Benioff plane (Hayes 2018). The seismic density distribution is shown in Figure 3A, panel A, we can
see that seismicity tends to cluster in the vicinity of the seaward projection of the TLF.

**2.5     Distance from the trench to the shelf brake**
Saillard et al. (2016), show that peninsulas along subduction zones cost lines present a long-term permanent
coastal uplift that can be associated with creep and aseismic slip domains. Thus, distance from the trench to
the coast (DTC) constitutes a proxy to separate seismotectonic segmentation due to the weak plate coupling.
The physics behind this proposal lies in the dragging force that subduction force induces on the overriding
plate, thus with less traction (weak plate coupling in the long term), the fore-arc region close to the trench
should be shallower than the surroundings. To gain a broader perspective of the peninsula's distribution,
Figure 3B contours the distance to the shelf brake, which is probably a better proxy for a potential uplifted
domain in the coastal region. As shown in this figure, the DTC presents variations along the trench. We
identify domains of short DTC associated with peninsulas in the region near to: Arauco; Valparaíso; Tongoy;
Punta de Choros; and Mejillones. Based on geological and geochronological evidence in three of these
peninsulas (Mejillones, Tongoy, and Arauco), Saillard et al. (2016) determined uplift rates in the range of
0.6–2 meters per thousand years in the associated terraces. These terraces have been continuously uplifting for
at least the last 0-5–0.8 Myr, indicating a long-term process compared to the seismic cycle of less than 500
years. Using this evidence, in addition to the inter-seismic GPS coupling, Saillard et al. (2016) infer that these
peninsula zones are associated with weak plate coupling where deformation is mostly accommodated by
creep. Again, qualitatively speaking, there is a tendency to find peninsula distribution where TLF tend to
concentrate in the coastal region.

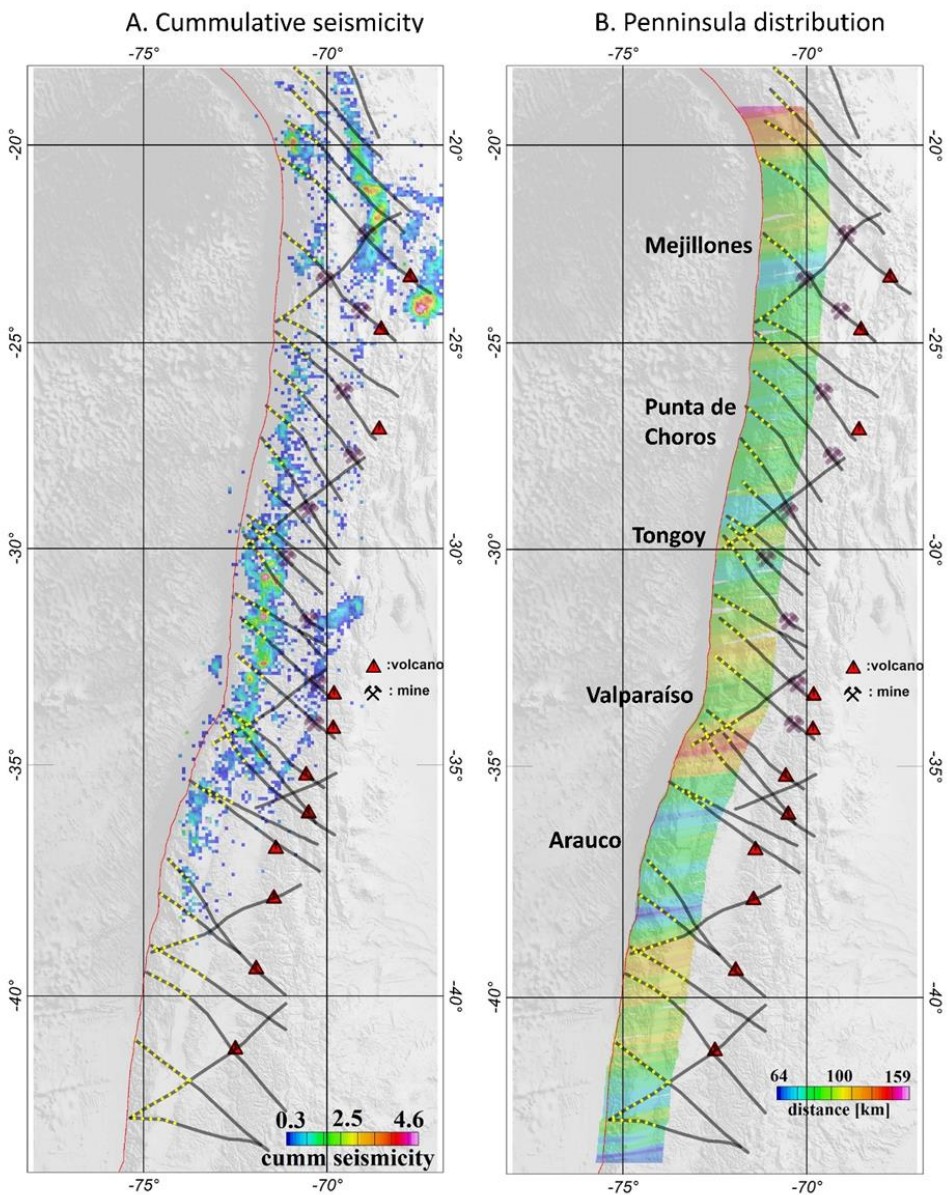

**Figure 3: Panel A: density distribution of the last 20 years of seismicity in the margin; values are normalized to better define the zones where seismicity has been concentrated, filtering out all the aftershocks associated with major megathrust activity (Taltal 2001, Maule 2010, Iquique 2014, and Vallenar 2015). Panel B: distance from the trench to the shelf brake, projected to the convergence direction (10E).**

## 2.6 Viscous coupling

The negative free-air anomaly along the Chile-Perú Trench is the response to dynamic equilibrium between

buoyancy and tectonic forces (Yáñez and Cembrano, 2014). The tectonic force tends to drag the continental





plate downwards, whereas buoyancy restores this deformation. Assuming equilibrium between the net
tectonic force and the long-term deformation (flow in continuum physics) the observed bathymetry represents
this force equilibrium. Therefore, for each bathymetric observation, a given Slip Layer Viscosity (SLV)
(Wdowinski, 1992) allows a match between observations and long-term viscous plate coupling. Using the
methodology developed by Yáñez and Cembrano (2004), we determine the along-strike SLV in the Nazca-
South America plate convergence region, considering across-strike profiles every 20 kilometres. As indicated
earlier, zones of maximum slip involve wavelengths larger than 20 kilometres for the megathrust events, and
therefore, a sample interval of 20 kilometres ensures appropriate along-strike resolution. In addition,
following the same rationale and conclusions of Yáñez and Cembrano (2004), we estimate that the increase of
the SLV in the north of the study area is due to a temperature-dependent rheology. This increase in viscous
plate coupling in the north is likely to be responsible for the larger crustal shortening observed in the southern
Andes in the last 20 Ma. Although other authors (e.g. Lamb and Davies 2003) consider that the deficiency in
sedimentary supply in the trench in the northern Andes is the driving mechanism for the larger viscous plate
coupling in the region. However, this discussion is beyond the scope of the present work, and since the
viscous plate coupling correctly represents the observations we are interested here in the short-wavelength
viscous plate response as a potential tool to identify zones with different degrees of coupling along the
convergent margin. Therefore, we remove this regional viscous plate coupling to isolate short-wavelength
features. This residual slip layer viscosity (RSLV) is included in Figure 4A (see a full discussion in
Supplement A). This signal shows positive (high relative viscous plate coupling) and negative (weak relative
coupling) zones. Again, we use normalized values to highlight the spatial distribution of the signal. In the
supplementary material, we present a full description of the modelling used to obtain the RSLV signal. As the
modelling is 1D, we extend the result of each model along the strike of the convergence (10°E).
**2.7      Inter-seismic GPS coupling**
GPS data provide information on the surface deformation relative to a stable continental reference. During the
inter-seismic period, the slip velocity at the intraplate contact, Vinter-seismic, can be determined from a GPS
network under the assumption of elastic plate deformation (e.g. Okada, 1985). This inter-seismic velocity
depends on the degree of plate coupling, $\phi$. At maximum plate coupling ($\phi$=1), Vinter-seismic is null, and at
minimum plate coupling ($\phi$=0), Vinter-seismic is equal to the convergence velocity (Vconvergence) (e.g.
Nuvel1a, De Mets, et al., 1994). Or, mathematically (e.g. Metois et al., 2012), Vinter-seismic=(1-$\phi$)*
Vconvergence. Inter-seismic GPS coupling is presented as GPS locking data in Panel B of Figure 4 (based in
a compilation of GPS information derived from different sources, Burgmann et al., 2005; Chlieh et al., 2008;
Loveless & Meade, 2011; McCaffrey et al., 2002; Metois et al., 2012, 2016; Moreno et al., 2010, 2012;
Wallace et al., 2004)). From 27°S to the north, high GPS plate coupling is generally observed, although some
correspondence is observed with the local minimum and TLF distribution. Between 27°–33°S, the GPS
coupling shows domains with lower values with better correspondence with TLF segmentation and the
minimum in viscous coupling. To the south of 33°S, the GPS plate coupling shows a spatial distribution that
again shows some coincidence with the other proxies, but also some discrepancies.   This is not surprising,
since GPS inter-seismic plate coupling reflects the quasi-instantaneous coupling of seismo-tectonic segments





at different loading stages. Nevertheless, in most of the studied segments, the GPS plate coupling correlates
relatively well with the viscous plate coupling, and the location of peninsulas and cumulative seismicity in the
last 20 years.

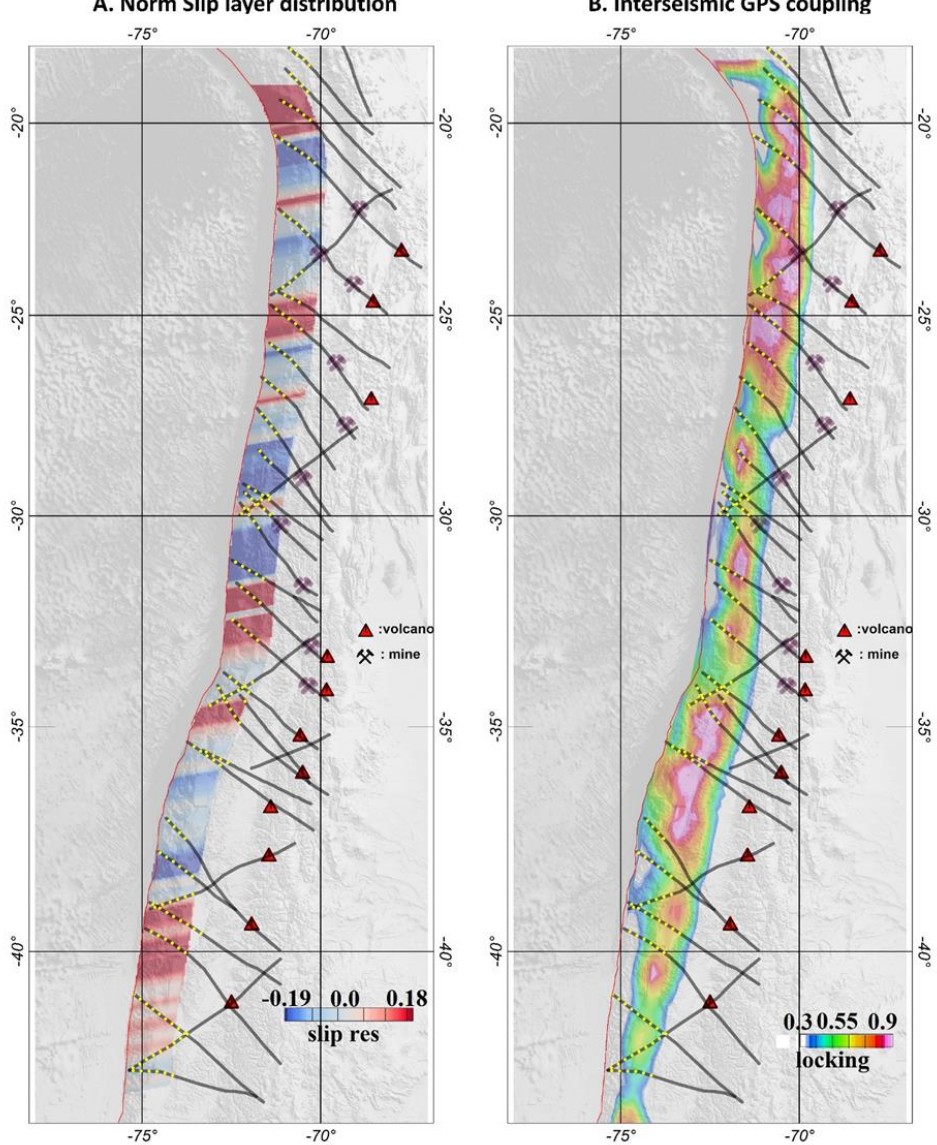


**Figure 4: Panel A: Normalized Residual slip layer viscosity (RSLV) derived from 1D modelling along profiles**
**separated every 10 km and oriented along the Nazca-South American plate convergence (10ºN); as this model**
**involves all of the slip layer, its spatial distribution is represented from the trench until 150 km landward. Panel B:**
**GPS inter-seismic plate coupling, model 2017 (Burgmann et al., 2005; Chlieh et al., 2008; Loveless & Meade, 2011;**
**McCaffrey et al., 2002; Metois et al., 2012, 2016; Moreno et al., 2010, 2012; Wallace et al., 2004).**





**3. Discussion**

**3.1 Quantitative correlation between TLF and plate coupling proxies derived from seismicity distribution, GPS and viscous coupling and coastal morphology.**

In order to better quantify the correspondence between the spatial distribution of TLF and the indirect estimate of plate coupling described in chapter 3 we present here an objective comparison between them. This task is challenging, taking into consideration the poorly constrained data used: (a) in some cases, regional-scale geological observations (TLF and peninsula distribution); (b) different time-scale coupling estimates (inter seismic GPS locking and long term viscous coupling); (c) poorly resolved GPS solution offshore; (d) 1D modelling of viscous coupling;   and (e) The lack of completeness in the seismicity record (historical record of 500 years, instrumental record of megathrust events of 50 years, and cumulative seismicity of 20 years) considering a seismic cycle of a couple of hundred years in the margin. Thus, none independent proxy is capable to produce a reliable estimate by itself, but rather a combination of them. Therefore, a thorough analysis is beyond the capabilities of the data source, and what we present here, though quantitative, should be understood as a guide to determine tendencies from different and independent perspectives that as a whole, provide a more robust estimate on the link between TLF and plate coupling in the margin.

The approach adopted considered the spatial correlation between TLF and the six proxies described in chapter 3, using the Pearson correlation coefficient between two variables (rxy) defined as:

$$r_{xy} = \frac{\sum_{i=1}^{n}(x_i - \bar{x})(y_i - \bar{y})}{\sqrt{\sum_{i=1}^{n}(x_i - \bar{x})^2 \sum_{i=1}^{n}(y_i - \bar{y})^2}} \qquad (1)$$

Where $\bar{x}$ and $\bar{y}$ are the average value of each variable. This function $r_{xy}$ has values between -1 (opposite correlation) to 1 (direct correlation). Values near zero mean weak or null correlation. In the application of the Pearson correlation in this case, the spatial distribution of TLF is always the $x_i$, and the 6 proxies used in this case are the $y_i$ in each case. A key property of the Pearson correlation coefficient is its invariance to spatial distribution of samples and scale of the two variables. This property is particularly useful in this case where we are trying to correlate very different proxies in terms of spatial distribution and scale.  The correlation is performed in moving windows bins of 32x32 km$^2$, with an overlap of 50% between correlation estimate. The correlation is calculated in a domain of 140 km width from the trench to the east, the plate coupling zone where short-term and long-term processes take place.

TLF are defined as line traces, but in order to spatially correlate them with the other variables, we add a width, considering potential spatial uncertainties and zones of influence. Thus, the width of each TLF is treated as a gaussian with a value of 1 in the centre and 0 at the edge, located at 10 km from the centre, representing the deformation zone and the lateral surface covered by the potential fluid release. Such a width of 20 km seems a reasonable number for a fault system of more than 100km length (>20%). In fact, in recorded earthquakes, like the Landers earthquake 1992 (Mw 7.3) where a rupture length of 85 km has been determined, with a shear deformation zone of 12-16 km (Perrin et al., 2020). Outside the TLF domains a value of -1 indicate no spatial distribution of TLF, but in practice is not relevant because the correlation is focussed inside the TLF domain



only. The other six proxies are treated in different manner, depending on its nature. GPS plate coupling is a spatial variable covering the whole spatial range of the coupling. Looking at the GPS coupling described in Figure 4b, we can see that most of the plate contact is highly coupled, well above 0.6 almost everywhere, thus in order to identify some differences in coupling we setup the mean value at 0.8. Slip viscosity layer and distance from the shelf brake to the trench are single values varying with latitude which are extended to spatial variables projecting the value landward following the convergence direction (~10°E). In the case of the slip coupling a mean value is already removed, thus a mean value of 0 is considered. For the shelf brake-trench distance we use the average separation of 100km as the mean value. Seismic cumulative density and slip distribution of megathrust events define restrictive domains along the plate coupling region. These areas are normalized between 1 and zero, and outside the region a value of -1 is assigned (no data). The same procedure is used for the boundary between historic seismicity segmentation, value 1 in the transition, and -1 outside. Since the analysis is restricted to the correlation between TLF's and the six proxies, the correlation only concerns the inner part of the TLF. Given the nature of each proxy, a low coupling at a given TLF implies a negative Pearson correlation at GPS and viscous coupling, distance from the shelf brake to the trench, and slip distribution for megathrust events (maximum slip should lie outside the TLF domain). On the contrary, positive Pearson correlation is expected with the historic segmentation and cumulative seismicity, to reflect low coupling at the TLF domain.

The results for each Pearson correlation coefficient spatial distribution are presented in Figure 5 in a plan view. In Figure 6 we present the result for the 32 relevant TLF in terms of the histogram obtained for the Pearson correlation inside the corresponding TLF domain. Over the histograms observations we include an interpretation on the correspondence with a low plate coupling condition, depending on the shape of the histogram, positive (Pearson correlation biased to the left in GPS, VISC, DIST, SLIP histograms; and biased to the right in the CUMM and HIS histograms), unclear (flat for all the proxies) and negative (Pearson correlation biased to the right in GPS, VISC, DIST, SLIP histograms; and biased to the left in the CUMM and HIS histograms) correlation. Based on this analysis we qualify the potential of each TLF in terms of its barrier potential, high, ambiguous, and poor. The criteria to stablish this qualification considers the following: (a) high potential: at most one correlation is negative and by majority are positive correlation; (b) ambiguous: at most two correlations are negative and at least one correlation is positive; (c) poor: when more than three correlations are negative or none of them are positive.

Some relevant conclusions arise from the spatial analysis of Figure 5 and histograms of Figure 6:

1. From the 32 relevant TLF in terms of plate coupling, 63% (20 of 32) show a high potential for a barrier behaviour, 31% (10 of 32) presents some ambiguity, and only 6% (2 of 32) TLF show a poor chance to become a barrier domain.

2. For the case of ambiguous potential, almost all of them present at least 2 positive correlation proxies.

3. For individual histograms, 54% histograms show a positive correlation, 28% are considered ambiguous, and 18% present a negative correlation.



4.  Five out of seven seismotectonic boundary segments present a strong correlation with TLF spatial distribution (Figure 5a). In terms of particular histogram distribution, 11 out of 13 show a positive correlation and none of them show a negative correlation.

5.  The cumulative interseismic seismicity (Figure 5a), perhaps the weakest proxy due to the lack of seismic completeness due to the very restricted time window of observation, still shows an almost 100% direct correlation with the TLF traces where inter-seismic activity developed (TLF 14-15-16, TLF 18-20, TLF 22). In terms of the histogram distribution, shows a rather similar pattern, some clear positive correlation in 6 out 18 TLF and a none conclusive solution in 12 out 18 cases.

6.  The spatial distribution of the slip zones of the main megathrust events recorded in the last 50 years, show a minimum positive correlation with the spatial distribution of TLF. As we can see in Figure 5b, less than 20% of the total slip domains, potential zones of asperities, correlate positively with TLF. The remaining 80% lies outside the zone of influence of TLF. In the histogram distribution, the same pattern is observed, 57% of negative Pearson correlation (or positive correlation in terms of low plate coupling), 26% of ambiguous solution and 17% positive Pearson correlation. It is important to note that in several histograms of this proxy a positive correlation is adopted when a low flat response is observed, but in the left side there is a single column saturated at the maximum value for correlation -1 (most of the TLF is empty, or in other words the slip zone lies outside the TLF domain).

7.  In the GPS plate coupling-TLF Pearson correlation coefficient (Figure 5b) 50% of the cases show a negative correlation (low relative coupling), whereas 30% show some mix results, with the negative correlation concentrated in the deeper parts of the coupling, and only in 20% of the cases a positive correlation holds, mostly concentrated in the coupling zone of the Antofagasta 1995 and Tocopilla 2007 earthquakes, and probably linked with some post seismic effects. Consistently, in 18 out of 32 (56%) histograms responses (Figure 6), the low coupling correlation is observed, whereas in 10 out of 32 (31%) the response is ambiguous, and the remaining 13% is associated with relatively high GPS coupling. We acknowledge that these values are very much conditioned by the choice of the threshold of 80% to separate high to low GPS coupling, but the aim is to identify less coupled domains in a signal almost saturated with high values.

8.  The same type of analysis was performed for the Slip Layer viscosity – TLF Pearson correlation coefficient (Figure 5c). In 50% of the case the correlation is opposite (low viscosity slip zones corresponds with the location of TLF). In 15% of the cases, we observed mixed results, whereas in 35% of the cases the correspondence is positive. Similar results are obtained with histogram responses (Figure 6), in 17 out of 32 (53%) the low coupling correlation is observed, whereas in 6 out of 32 (19%) the response is ambiguous, and the remaining 28% is associated with relative high slip viscosity. One important limitation of this approach is the 1D approximation of an inherently 3D process. This fact is probably the main reason for its relatively low positive response compared



to the other proxies. Finally, figure 5c show the Pearson correlation coefficient for the distance from
the shelf brake to the trench. In this case, the closest shelf brake to the trench at TLF intersection is
a 36%, the same number of cases show an opposite behaviour and only 28% presenting mixed
results. In terms of the histogram distribution (Figure 6), the same tendency is observed, but with a
higher predominance of shorter distance shelf brake-trench (44%), whereas the opposite is observed
in 34% of the cases and 22% show an ambiguous response. This is the proxy that show the lowest
level of positiveness, probably due to the fact that other processes are also involved in the uplift of
the peninsula regions, for instance the density of the crust and its relative buoyancy.
As we point out earlier in the text, none of the proxies by itself have the merit to account for the degree of
coupling along the subduction zone, and the results emanated from the Pearson correlation demonstrate that.
However, when we integrated the individual results 63% of the TLF can potentially behave as barrier, and only
in two cases (6%) chances are poor. In the remaining 31% of the cases, represented as ambiguous cases, there
are still some evidences of positive correlations in more than one proxy. In Figure 5a, panel A, we include a
reference for the TLF with high potential to become a barrier (green arrow), and we can see that in almost all
the cases they are consistent with the tectonic segmentation derived from the historic seismicity. One peculiar
distribution of potentially active barrier domains is observed between 25°-30° S, the zone with less historical
seismicity (Figure 2). On the other hand, not necessarily all the TLF behave as barriers, due to lack of favourable
orientation, depth extent, age, dip angle, fluid content among other uncertainties. Therefore, we consider that
the previous semiquantitative analysis including all the proxies, support the presence of a geological signal of
low plate coupling when TLF is present.  In the next section we propose a conceptual mechanism to explain
this phenomenon.



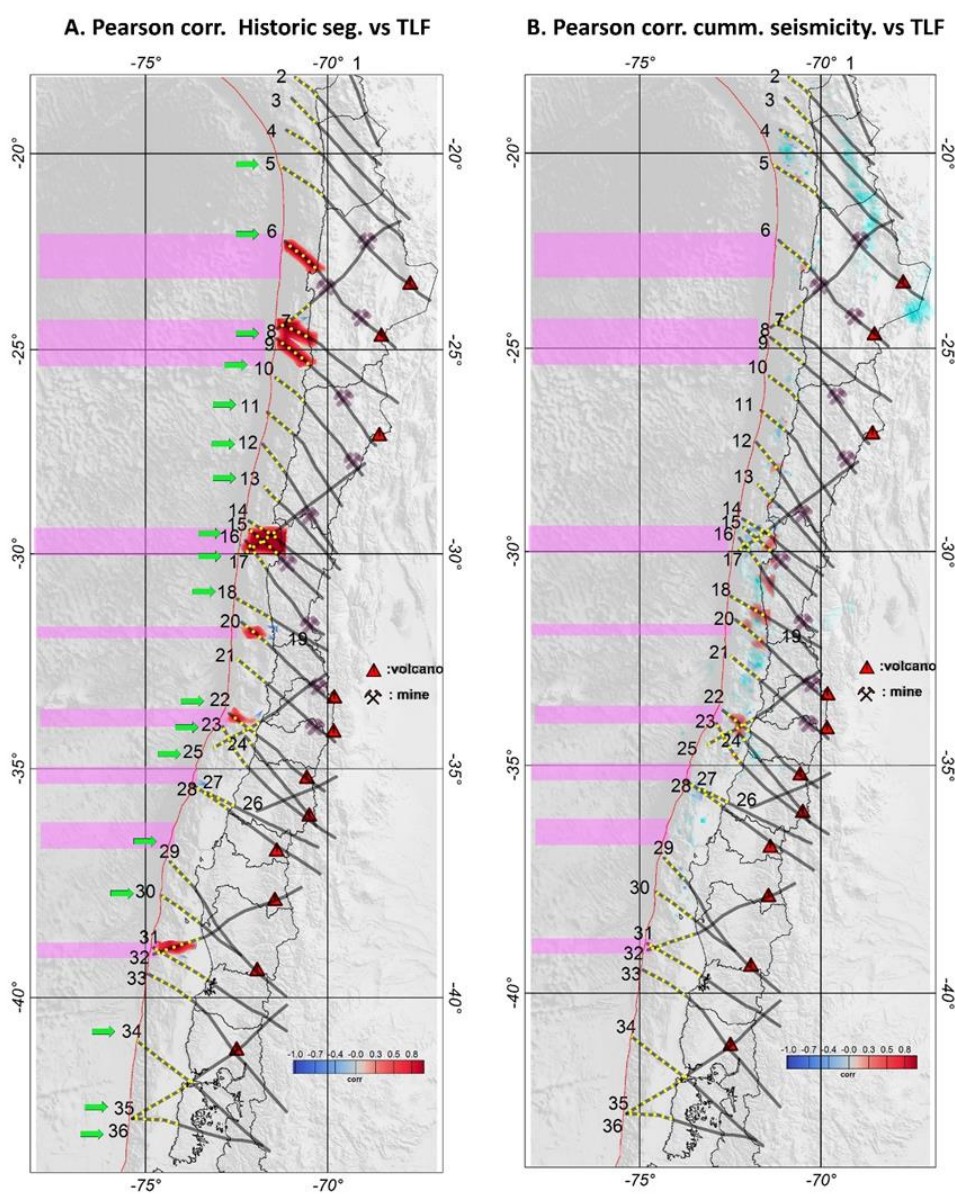

**Figure 5a: Pearson correlation coefficient between TLF and (A) tectonic segmentation and (B) cumulative seismicity. Colour code range from -1 (opposite correlation) to 1 (direct correlation). In panel A the green arrow shows the TLF with high potential as a barrier, according with the criteria stablished from histograms distribution of Figure 6.**

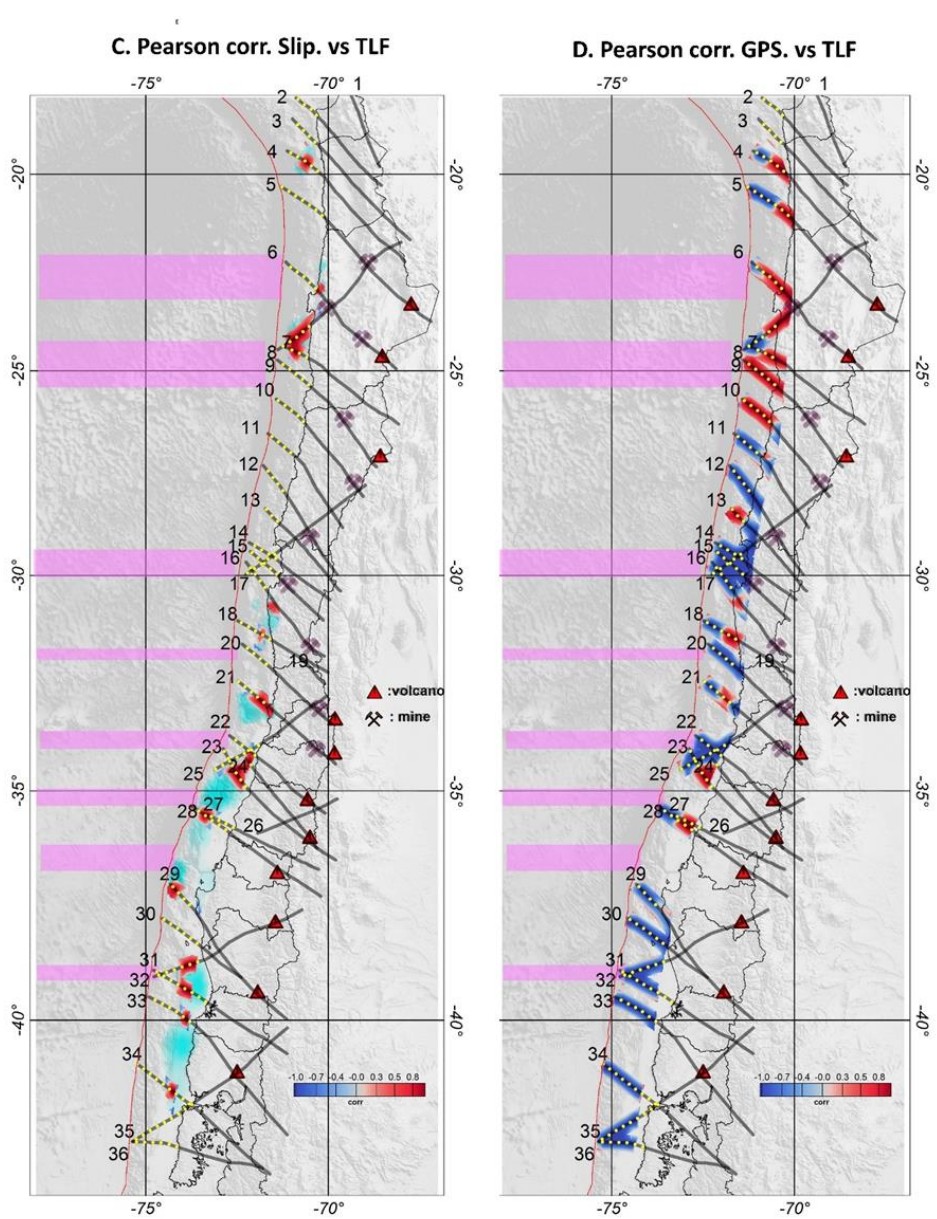

**Figure 5b: Pearson correlation coefficient between TLF and (c) normalized slip viscosity and (d) GPS coupling.**
**Colour code range from -1 (opposite correlation) to 1 (direct correlation).**





**Figure 5c: Pearson correlation coefficient between TLF and tectonic slip viscosity (e), and distance from the trench to the shelf brake (f). Colour code range from -1 (opposite correlation) to 1 (direct correlation).**



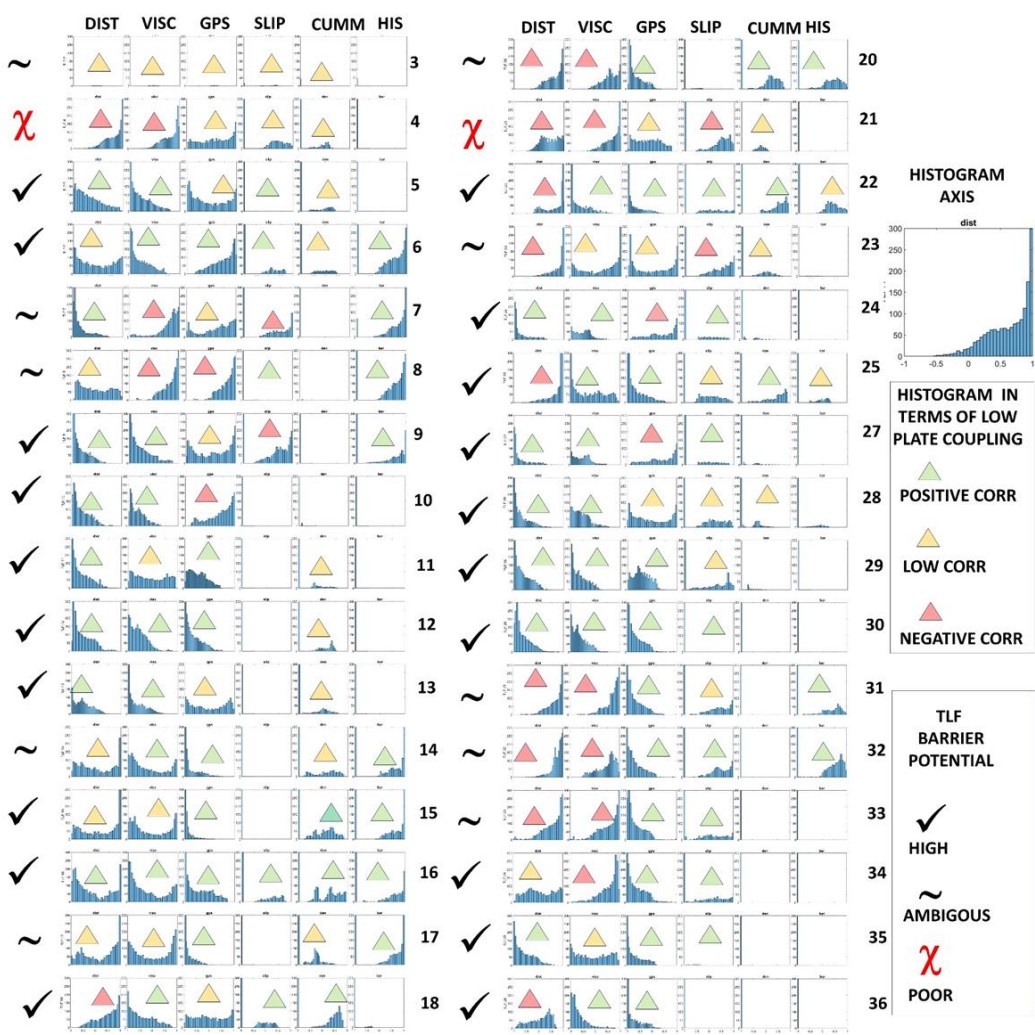

**Figure 6: Histogram diagrams for Pearson correlation in each TLF. Histogram interpretation and TLF qualification as a potential barrier is indicated in inlet. TLF number is indicated to the left of each panel (a good resolution of this image is provided in the supplementary material).**





**3.2 A simple conceptual barrier model: misoriented TLF as a store/released of fluids during the seismic cycle.**

Comparing the spatial distribution of the seaward extension of the TLF and the previously discussed first-order conditioning factors of the tectonic segmentation in the Andes (chapter 2), and the cross correlation described in section 4.1, we can make the following conclusions:

1. The coastal termination of an TLF generally occurs close to a peninsula, where the shortest trench–coast distance is observed, in spatial correspondence with zones of negative RSLV (weak viscous coupling), and in some cases also corresponding to zones of weak GPS coupling. However, it should be noted that the degrees of coupling inferred via RSLV and GPS do not map similar observation time windows, covering geological (Ma) vs seismic cycle (300–500 years) time frames, respectively.

2. During the last 60 years, slip displacements during the major megathrust earthquakes in the margin of Chile tend to be bounded by the coastal termination of an TLF in their northern and southern boundaries. Thus, if these slip zones represent a spatial distribution of asperities, the TLF correspond to zones potentially associated with barriers, consistent with the long-term low coupling inferred from RLSV, GPS plate coupling and distributions of peninsulas. the previous long-term observations.

3. Cumulative seismic activity in the last 20 years tends to nucleate in the vicinity of the seaward termination of the TLF, normally with the development of seismic swarms of 100–300 events of medium to low magnitudes during periods of several weeks at most.

4. The geological record demonstrates that TLF are long-lived structures of high permeability in comparison with the surrounding crust and most likely the underlying mantle as well, and are thus potentially efficient fluid storage structures.

The previous observations provide the grounds to propose a simple conceptual model to understand the role played by TLF in the tectonic segmentation of a convergent margin. These observations consistently show that TLF in the seismogenic zone are spatially correlated with long-term and short-term evidence of weak coupling behaviour. The long-term evidence involves geological processes that build up during many seismic cycles, over a time frame of millions of years, including low values of slip-layer viscosities and correspondence with the spatial distribution of peninsulas. The short-term evidence involves fragments of the seismic cycle over a time frame of less than 500 years, characterized by weak coupling zones as inferred by inter-seismic GPS observations, the flanks of slip zones of recent mega-thrust events, and the boundaries that delimit the historical record of major events. Overall, these observations consistently show that TLF domains are likely candidates for barrier zones.

From basic principles, the strength of a fault is controlled by the friction at the discontinuity plane. According to Amonton's law, the fault strength is proportional to the product of the normal stress and the static or dynamic friction (e.g. Scholz, 1990). In the presence of fluids, pore pressure reduces the normal stress, thereby reducing the strength of the fault (e.g. Scholz, 1990), eventually to zero if the pore pressure reaches the lithostatic pore



pressure. Under these supra-lithostatic fluid pressure conditions, even faults that are strongly misoriented for
frictional reactivation under the prevailing stress field can be reactivated, focusing the discharge of large
amounts of overpressured fluids and acting as a "fault-valve" (Sibson, 1990; Cox, 2016). Indeed, Cox (2016)
showed that, under supra-lithostatic fluid pressure conditions, the typical seismic response in the faults
corresponds to microseismic swarms, which, according to Sibson (2020), would concentrate at the roots of the
fault system. In the case of an TLF, which is a long-lived structure transecting the whole lithosphere (e.g., Lutz
et al., 2022), the root of the fault system at the Andean convergent margin corresponds to the subduction
channel. Low fault strength at subduction zones can be equated to barrier zones where convergence is mostly
accommodated by creep and/or micro-seismicity. The hydration of the subducting slab during its bending in the
outer rise region has been widely documented in different subduction margins (e.g., Holbrook et al., 1999;
Shillington et al., 2015; Contreras-Reyes et al., 2007; Moscoso and Grevemeyer, 2015; Ranero and
Sallarès, 2004; Fujie et al., 2018, among others), as has the slab's subsequent dehydration during subduction
(Barriga et al., 1992; Maekawa et al., 1993; Peacock, 1993). Mantle hydrous phases (serpentinites) have also
been observed in forearc regions at subduction margins (e.g. Hyndman and Peacock, 2003; Xia et al., 2015;
Hansen et al., 2016), further demonstrating that subduction systems transport large amounts of water; however,
the amount of water transported is still unknown (Miller et al. 2022). On the other hand, fluid flow in porous
media is governed by Darcy's law, in the opposite direction to the hydraulic head and proportional to the
hydraulic permeability. Numerical models (Menant et al., 2019) have been used to determine the path of
overpressured fluid flow along the subduction channel, and how strong/weak frictional channels condition the
flow (weak frictional channel zones percolate more water upwards compared to strong frictional channel zones).
These two domains determine the location of weak and strong coupling zones at the plate contact. Thus,
according to basic principles and numerical models, water concentrates in zones of high permeability.
The geological record on land shows that, in the Andean margin, TLF are associated with ore deposits
clustered at the intersection of magmatic arcs that become progressively younger eastward (Piquer et al.,
2016; Yáñez and Rivera, 2019; Piquer et al., 2021a; Farrar et al., 2023; Wiemer et al., 2023), covering the full
tectono-magmatic history during the Mesozoic and Cenozoic. Local seismic networks deployed in Northern
and Central Chile also show alignments of seismic activity along some TLF systems (Yáñez and Rivera,
2019; Piquer et al., 2019, 2021a; Sielfeld et al, 2019; Pearce et al., 2021). These long- and short-term
observations indicate the presence of long-lived high-permeability domains along the TLF systems in the
Andean margin of Northern and Central Chile. Therefore, we postulate that TLF act as fluid sinks in the
forearc region, following a continental-scale fault-valve behaviour, carrying the fluids released by slab
dehydration and transported from distal locations through the subduction channel and discharging the fluids
upwards and laterally through the TLF. Thus, if the proposed mechanism operates for long periods of time,
the fluid distribution at the plate contact should show an uneven distribution of fluid, delimitating domains of
weak and strong friction channels, which would act as seismic barriers and asperities, respectively. In this
context, the spatial distribution of TLF would be associated with barriers that delimit the tectonic
segmentation. In the proposed model, tremor or swarm seismic activity represent episodic fluid release from
TLF that are poorly oriented with respect to the regional tectonic stress — in this case, the NW-striking fault



systems oriented at a high angle relative to the ENE convergence direction. This model provides a causal link
between the presence of TLF in the upper plate and the distribution of barrier and asperity domains in the
plate interface. A schematic cartoon of this model is presented in Figure 7.

.

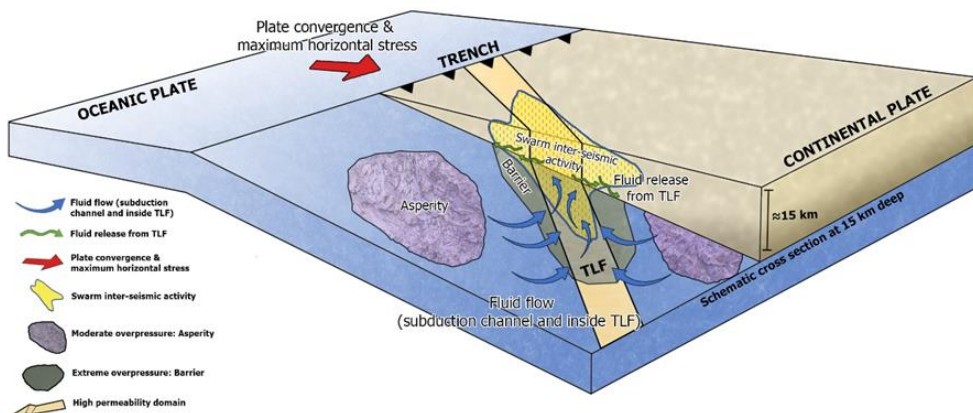


**Figure 7: Schematic conceptual model of fluid transport towards TLF, following different paths in the subduction channel, as well as upwards within the TLF. This model proposes that TLF are sink domains of slab-derived fluids that promote the development of barrier zones and dry out the neighbouring domains where asperities develop. Swarm clustering in spatial association with the TLF represents a mechanism for the quasi-creep release of energy within the barrier zone.**

### 3.3 Implications

If TLF act as low-friction domains (barriers) due to their capacity to store fluids released from the subducting
slab and thereby dry out neighbouring zones of the subduction channel, promoting the development of a high-
friction domain (asperity), we can envision a series of implications derived from the proposed model.
The most relevant implication is the geological control of barrier zones. This geological control exerted by high-
permeability domains in the continental lithosphere (TLF) implies a spatial control of barrier zones, and thus
the seismotectonic segmentation should be stable for several seismic cycles as long as the capacity of TLF to
store fluids is maintained. If this scenario is correct, the estimate of the seismic risk associated with each
seismotectonic segment can be assessed based on empirical fault-length laws (e.g. Anderson et al., 2016). In
this context, interplate seismic swarms and slow seismic events that develop in the vicinity of TLF zones would
be a mechanism for the steady release of seismic energy.
As discussed previously, several TLF have been identified in the Andean margin; however, little is known about
their origin, width, dip, depth extent, and capacity to behave as a water sink. Therefore, further study is needed
to postulate a reliable map of barrier domains in this subduction system.



On the other hand, seismic barriers/asperities would be conditioned by the capacity of barrier zones to
mobilise and store fluids, and would thus be relatively stable in space but with a variable behaviour during
several seismic cycles. If the age of the subducted slab conditions the water budget at the plate interface
(Rupke et al., 2004), the progressive age increase from south to north in this margin (from 0 to 45 Ma) would
be a controlling factor for the efficiency of the TLF-barrier hypothesis. Although this implication is highly
speculative, the historical record shows that the largest megathrust events at the margin have occurred in
Southern Chile, including the 9.3Mw 1960 Valdivia Earthquake, the largest event recorded worldwide.

4.   **Conclusions**
Based on first order geological and geophysical observations of the Nazca-South America plate convergence
we propose a conceptual model to understand the tectonic segmentation in the Andean region.
Observations include historical seismicity and the associated seismotectonic segmentation. Major thrust events
occurred in the region in the last 60 years, defining domains of asperities. GPS and viscous plate coupling that
provide independent proxies to stablish potential domains of barriers (low plate coupling) and asperities (high
plate coupling). Location of low plate coupling domains is further associated with the spatial distribution of
peninsulas (less basal erosion) and cumulative seismicity during the inter-seismic period (slow interplate
seismic events, creeping, associated with fluid release).
Key element in the model is played by trans-lithospheric faults (TLF). Landward, this TLF system concentrate
the occurrence of major hydrothermal ore deposits and some active volcanism, denoting their intrinsic high
permeability. Thus, at their seaward edge the TLF domains act as sink and release of fluids during the seismic
cycle. The fluid is captured from the slab through the subducting channel, and continuously release to the plate
contact, promoting the growth of barriers beneath them (excess of fluids), and asperities laterally (reduction in
fluid content).
If the interaction of first order continental structures and the fluid content of the subducting slab plays a
central role in the seismotectonic segmentation of convergence zones, a carefully understanding of the
overriding plate geology and associated structures could be instrumental to better understand the associated
seismic risk.

**Competing interests:** The contact author has declared that none of the authors has any competing interests.
**Acknowledge**: This research was partially supported by Fondef project D10I1027. J.P. acknowledges support
from ANID-FONDECYT grant 11181048 and Amira Global P1249 project.
**Data Availability Statement**: The data used in this paper is derived from published papers, indicated in the
text, and topographic/bathymetric data extracted from public source, ETOPO 2022. DOI: 10.25921/fd45-gt74.



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
