# Peer review of "On the role of trans-lithospheric faults in the long-term seismotectonic segmentation of active margins: a case study in the Andes"

_EGUsphere, 2024_

## Referee Comment (RC1)

**Reviewer's comments on the article called:**

**On the role of trans-lithospheric Faults in the long-term seismotectonic segmentation of active margins: a case study in the Andes.**

**General comments**

The paper entitled: On the role of Trans-Lithospheric Faults in the long-term seismotectonic segmentation of active margins: a case study in the Andes by the authors Gonzalo Yáñez, José Piquer and Orlando Rivera, seeks to establish the hypothesis that the large structures called Trans-Lithospheric Faults recognized in the active continental margin of Chile, could have an influence in the seismotectonic segmentation of large subduction earthquake ruptures, because these structures would be able to transport and contribute an important amount of fluids to the subduction zone, producing a creeping zone surrounded in a more coupled zone. To prove this, the authors establish spatial relationships with different observations and factors determined at the margin among them are: historical seismicity, distance between the trench and the continent, coupling models and Pearson correlation parameters. Although it is a novel hypothesis and the manuscript is clear and well written, there are certain aspects that are not clear to me both in the writing, the postulated and the Figures presented that in my opinion are necessary and I request to improve the article. These aspects are specified below.

**Specific comments**

In lines 106-110 of the manuscript, it is explained how Trans-Lithospheric Faults (TLF) have been defined through several observations. One of these aspects you point out is the seismicity associated with this type of structures, with which we could have an idea of the depth that these structures reach. However, I am very surprised that in Figure 1 (introductory) none of the TLFs have associated seismicity. This is why I ask that in Figure 1 they incorporate a panel B showing the cortical seismicity associated with this type of structures. In the manuscript they indicate that thanks to temporal networks it has been possible to detect seismicity, therefore, it seems to me relevant to incorporate in Figure 1 a panel B showing this seismicity. Showing this seismicity associated with these faults is something powerful that would undoubtedly help to improve the quality of the article.

On the other hand, a doubt: ¿are TLFs restricted in depth and spatially to the continental upper crust or can they also partly affect the oceanic crust? Please make this clear when introducing TLFs in the manuscript **(lines 106-110).**

2.- In Figure 2, it strikes me that the Iquique 2014, Tocopilla 2007 and Antofagasta 1995 earthquakes do not follow the hypothesis put forward in the article. In these earthquakes the zone of greater slip or roughness, is just located in the trace of the TLF recognized in this place and not so in the earthquakes of the south, where if the postulated by you in the article

is fulfilled, ¿how can I explain this difference between the earthquakes of the north and the south with respect to your hypothesis? Please deepen this through a deeper discussion.

**Line 305:** although the coupling models indicated are good, there are new models published especially in the segment between Antofagasta and Copiapo. I recommend perhaps updating the models of this article with the most recent models published and incorporating to the references of these articles: Yáñez-Cuadra et al., 2022 (Geophysical Research Letters) and González-Vidal et al., 2023 (Geophysical Research Letters).

In **lines 453-458** it is explained that at 25° and 30°S there is a potential barrier zone due to the high correlation of the Pearson index. However, these zones also coincide with the Taltal ridge subduction at 25°S (León-Rios et al., 2024 G3) and the Challenger Fracture zone at 30°S (Poli et al., 2017 Geology; Maksymowicz, 2015 Tectonophysics). In that sense, further discussion of this correlation is lacking in the manuscript. Please discuss these points, as, while there is a spatial correlation between these barrier zones with TLFs, there is also correlation with other important bathymetric structures, which can either carry a significant amount of fluids or produce a considerable degree of fracturing, enhancing creeping seismogenic behavior. Incorporate a deeper discussion considering other possibilities to the correlations you find, i.e., incorporate to the article that, although you find a correlation between TLFs and creeping barrier zones, this would not be the only possibility. When improving this discussion, please incorporate the references mentioned above.

**Specific comments for Figures**

**Figure 2:**

In panel A, the symbology used of gray lines indicating magnitude is very confusing and not well understood. Although it may be useful for higher magnitude earthquakes, for magnitude 7 events the line is too thin and cannot be identified well in the Figure. On the other hand, the word magnitude is in Spanish and not in English.

The caption of the Figure is incomplete and is not in tune with what is written in the manuscript. The segmentation says that it is marked by semitransparent yellow ribbons when in fact they are pink.

In panel B, please point out to which earthquake (earthquake name) each slip patch corresponds. There may be readers who are not familiar with Chile's earthquakes, so indicating or pointing out each earthquake in the Figure (panel B) may be helpful to readers.

I recommend improving or rewriting the caption of this Figure to be more precise in the information provided.

**Figure 3:**

It is missing to indicate in the caption that the seismicity was extracted from the National Seismological Center.

I think there is an error in indicating the 2015 earthquake as "Vallenar 2015" in the caption, is it not the Illapel earthquake of 2015? I have no recollection of a Vallenar earthquake in that year.

Incorporate the abbreviation DTC in panel B, it could be indicated on the color scale indicating distance.

In general, I recommend rewriting or rephrasing all the captions of the Figures as well as the wording of these. As they are written they give very little information and are inaccurate. They could definitely be much better.

**Figure 7**

Enlarge the letters of the symbology

**Technical corrections**

**Line 23:** specify in a better way what type of observations are referred to, these can be seismotectonic, seismological, geodetic...etc.

**Line 44:** take out "including the development of asperities and barriers in the same spatial and time frame".

**Lines 49 to 51:** In this part it seems necessary to include Scholz's reference that indicates these different landslide states.

**Line 67:** add reference Moreno et al., 2014 Nature Geoscience.

**Line 81**: Hayes et al., 2018? Or just Hayes, 2018? In this publication it is not just Hayes, 2018, it is Hayes et al., 2018.

**Line 82:** Yanez to Yañez et al., 1988.

**Line 152:** Add reference Calle-Gardella et al., 2021 Journal of Seismology.

**Lines 196-199:** this sentence is confusing, please rewrite or rephrase.

**Line 209:** Vi to VI

**Line 219:** Magnitude Mw 9.3 What reference determines this magnitude? Please incorporate reference or change the magnitude.

**Line 237:** remove double parenthesis in "Omori's Law".

---

## Author Comment (AC3)

Dear Anonymous reviewer:

Many thanks for your thorough and dedicated review of our paper. We are sure that your observations contribute to a better explanation of the ideas behind this research contribution.

In the following we present our answers to your questions and observations. Every answer follows the specific question/observation, in bold italics, and a direct reference to the modifications made in the text.

Best regard,

Gonzalo Yanez

General comments

The paper entitled: On the role of Trans-Lithospheric Faults in the long-term seismotectonic segmentation of active margins: a case study in the Andes by the authors Gonzalo Yáñez, José Piquer and Orlando Rivera, seeks to establish the hypothesis that the large structures called Trans-Lithospheric Faults recognized in the active continental margin of Chile, could have an influence in the seismotectonic segmentation of large subduction earthquake ruptures, because these structures would be able to transport and contribute an important amount of fluids to the subduction zone, producing a creeping zone surrounded in a more coupled zone. To prove this, the authors establish spatial relationships with different observations and factors determined at the margin among them are: historical seismicity, distance between the trench and the continent, coupling models and Pearson correlation parameters. Although it is a novel hypothesis and the manuscript is clear and well written, there are certain aspects that are not clear to me both in the writing, the postulated and the Figures presented that in my opinion are necessary and I request to improve the article. These aspects are specified below.

Specific comments

In lines 106-110 of the manuscript, it is explained how Trans-Lithospheric Faults

(TLF) have been defined through several observations. One of these aspects you point out is the seismicity associated with this type of structures, with which we could have an idea of the depth that these structures reach. However, I am very surprised that in Figure 1 (introductory) none of the TLFs have associated seismicity. This is why I ask that in Figure 1 they incorporate a panel B showing the cortical seismicity associated with this type of structures. In the manuscript they indicate that thanks to temporal networks it has been possible to detect seismicity, therefore, it seems to me relevant to incorporate in Figure 1 a panel B showing this seismicity. Showing this seismicity associated with these faults is something powerful that would undoubtedly help to improve the quality of the article.

*We agree with the reviewer on the great value of having seismicity directly associated with TLF, but this is not the case, most likely due to their large recurrence time, in the time frame of thousand years . The focus of the paper is the seismicity in the subduction plane, in other papers, like Piquer et al., 2019, we discuss the few evidences of seismicity linked to TLF, but not enough evidences to populate a panel in Figure 1. Although indirect evidences of activity related with ETL is presented in Figure 3a, in the cumulative inter-seismic activity, in particular during seismic swarms, and the normal event of March 11, 2010 at the Pichilemu TLF (22) (linked to the Maule 8.8 Mw event of 2010).*

On the other hand, a doubt: ¿are TLFs restricted in depth and spatially to the continental upper crust or can they also partly affect the oceanic crust? Please make this clear when introducing TLFs in the manuscript (lines 106-110).

*We don't know in detail the TLF behaviour with depth, from the geological and geophysical evidences that show the alignment of magmatic and hydrothermal activity we are confident that they involve the whole lithosphere. We have no evidence of a prolongation towards the oceanic crust below the Benioff plane, most likely is not the case due to the creep nature of the process postulated for the interaction of TLF and plate coupling. We added a sentence in the paragraph to clarify this point:*

*"The geometry and depth extension of TLF is unknown, but based on their control of continental-scale magmatic and hydrothermal processes and their surface*

*traces in the order of hundreds of kms, we consider that they involve, exclusively, the whole lithosphere".*

2.- In Figure 2, it strikes me that the Iquique 2014, Tocopilla 2007 and Antofagasta 1995 earthquakes do not follow the hypothesis put forward in the article. In these earthquakes the zone of greater slip or roughness, is just located in the trace of the TLF recognized in this place and not so in the earthquakes of the south, where if the postulated by you in the article is fulfilled, ¿how can I explain this difference between the earthquakes of the north and the south with respect to your hypothesis? Please deepen this through a deeper discussion.

*We partially agree with the reviewer observation, for the case of Iquique 2014 event, Iquique TLF (4) is cutting the slip zone however the offshore extension of this TLF is not well resolved (in the seaward extrapolation we use bathymetric morphology as the principal guide), if it continues straight from the landward side, most of the slip zone would be to the south of TLF 4. For the case of Tocopilla 2007 half of the slip zone is outside the slip zone definition. Finally, for the case of the Antofagasta 1995 event, we totally agree with the reviewer observation, the slip zone is indeed cut by two TLF (7: Agua Verde-Exploradora, and 8: Antofagasta-Chonchi). Thus, in these particular cases against the model prediction, we envision two possible explanations for this lack of consistency: (1) the fact that this is a low magnitude event (8 Mw) compared to the other cases, and or (2) not all TLF behave as barriers. We add a discussion of this particular lack of consistency in point 6 of the discussion section 3.1 as follows*:

*"The most conspicuous case against the rule is the slip zone of the Antofagasta 1995 that cut two TLF (7: Agua Verde-Exploradora, and 8: Antofagasta-Chonchi) and partially the Tocopilla 2007 event (Mejillones-Llullaillaco TLF 6). Two complementary explanations are proposed in this case: (1) both are small events (8Mw) compared to the other megathrust events, (2) not necessarily all TLF behave as barriers all the time. For the case pf Iquique 2014 event, the seaward extension of of Iquique TLF is not well constrained, and most likely run straight from landward segment, leaving the slip zone entirely to the south of TLF 4.  ."*

Line 305: although the coupling models indicated are good, there are new models published especially in the segment between Antofagasta and Copiapo. I recommend perhaps updating the models of this article with the most recent

models published and incorporating to the references of these articles: Yáñez-Cuadra et al., 2022 (Geophysical Research Letters) and González-Vidal et al., 2023 (Geophysical Research Letters).

*Thanks for providing these new references. Looking at the new coupling models derived from GPS observations as shown in these two papers we noticed that results do not depart significantly with the model presented in Figure 4b, and for the large-scale purpose of our research is not adding more information, so we decided to keep the original GPS coupling. But we add a sentence in section 2.7, explaining that the new GPS models in the northern Chile region are consistent with the GPS model used in the paper:*

*"For the segment between Antofagasta and Copiapo (24-28°S), two new GPS plate coupling models are available (Yáñez-Cuadra et al., (2022) and González-Vidal et al., (2023)), however, we noticed that these new results share similarities with the model presented in Figure 4b, and is therefore not necessarily included in this case."*

In lines 453-458 it is explained that at 25° and 30°S there is a potential barrier zone due to the high correlation of the Pearson index. However, these zones also coincide with the Taltal ridge subduction at 25°S (León-Rios et al., 2024 G3) and the Challenger Fracture zone at 30°S (Poli et al., 2017 Geology; Maksymowicz, 2015 Tectonophysics). In that sense, further discussion of this correlation is lacking in the manuscript. Please discuss these points, as, while there is a spatial correlation between these barrier zones with TLFs, there is also correlation with other important bathymetric structures, which can either carry a significant amount of fluids or produce a considerable degree of fracturing, enhancing creeping seismogenic behaviour. Incorporate a deeper discussion considering other possibilities to the correlations you find, i.e., incorporate to the article that, although you find a correlation between TLFs and creeping barrier zones, this would not be the only possibility. When improving this discussion, please incorporate the references mentioned above.

*We acknowledge the fact that other features associated with the oceanic Nazca plate, like aseismic ridges, and fracture zones can carry large volumes of fluids that can also enhance the fluid pressure at the Wadatti-Benioff zone acting in complementary fashion with the proposed mechanism. We include a new paragraph at this regard in the discussion section 3.2.:*

*"Our proposed conceptual model in which TLF's promote the development of barrier domains along the subducting margin through the enhancement of fluid pressure complement other process at subduction zones that also enhances the budget of localized fluids at the plate contact, among them the collision of aseismic ridges and fracture zones, bending of the subducting plate (e.g. Ranero et al., 2008, Ranero et al., 2005, Martinez-Loriente et al., 2019; Arai et al., 2024). In the Nazca-South America plate interaction authors had highlighted this increase in fluids at passive ridges such as the Taltal ridge 33°S (Leon-Rios et al., 2014) and the Juan Fernandez ridge 33.5°S (Garrido et al., 2002), and fracture zones such as the Challenger Fracture zone 30°S (Poli et al., 2017; Maksymowicz, 2015). The volume of fluids in aseismic ridges is enhanced by oceanic water percolation along the thicker oceanic crust, while in fracture zones as a result of the high permeability that provides a mechanism to increase water storage prior to subduction.  These complementary mechanisms share a common origin at the subducting plate, and in the particular case of the Nazca plate they are oblique to the margin (roughly NE).  Thus, the main difference with the proposed model is their along strike migration with time, while in the proposed mechanism TLF belongs to the overriding plate."*

Specific comments for Figures

Figure 2:

In panel A, the symbology used of gray lines indicating magnitude is very confusing and not well understood. Although it may be useful for higher magnitude earthquakes, for magnitude 7 events the line is too thin and cannot be identified well in the Figure. On the other hand, the word magnitude is in Spanish and not in English.

*Figure 2a corrected and improved in terms of the visibility of small events (making to black the magnitude legend, and putting lighter the topo/bathymetry background)*

The caption of the Figure is incomplete and is not in tune with what is written in

the manuscript. The segmentation says that it is marked by semitransparent yellow ribbons when in fact they are pink.

***Caption corrected***

In panel B, please point out to which earthquake (earthquake name) each slip patch corresponds. There may be readers who are not familiar with Chile's earthquakes, so indicating or pointing out each earthquake in the Figure (panel B) may be helpful to readers.

***Included the names of the major events in panel B***

I recommend improving or rewriting the caption of this Figure to be more precise in the information provided.
***Caption redaction improved***

Figure 3:

It is missing to indicate in the caption that the seismicity was extracted from the National Seismological Center.
***Included***

I think there is an error in indicating the 2015 earthquake as "Vallenar 2015" in the caption, is it not the Illapel earthquake of 2015? I have no recollection of a Vallenar earthquake in that year.

***Modified***

Incorporate the abbreviation DTC in panel B, it could be indicated on the color scale indicating distance.
***Included***

In general, I recommend rewriting or rephrasing all the captions of the Figures as well as the wording of these. As they are written they give very little information and are inaccurate. They could definitely be much better.

***Most of the captions have been improved, with a more complete description of each figure panel.***

Figure 7

Enlarge the letters of the symbology

*Legend corrected*

Technical corrections

Line 23: specify in a better way what type of observations are referred to, these can be seismotectonic, seismological, geodetic...etc.
**To keep this sentence of the abstract succinct, we include the end members only:**

 **"We tested this hypothesis against key short- and long-term observations in the study area, seismological, geodetic, and geological, obtaining consistent results."**

Line 44: take out "including the development of asperities and barriers in the same spatial and time frame".
*Removed*

Lines 49 to 51: In this part it seems necessary to include Scholz's reference that indicates these different landslide states.
*Added*

Line 67: add reference Moreno et al., 2014 Nature Geoscience.
**Added**

Line 81: Hayes et al., 2018? Or just Hayes, 2018? In this publication it is not just Hayes, 2018, it is Hayes et al., 2018.
*The reference is indeed Hayes 2018:*

**Hayes, G. (2018). Slab2 - A Comprehensive Subduction Zone Geometry Model [Data set]. U.S. Geological Survey. https://doi.org/10.5066/F7PV6JNV**

Line 82: Yanez to Yañez et al., 1988.
*Corrected*

Line 152: Add reference Calle-Gardella et al., 2021 Journal of Seismology.
*Added, thanks*

Lines 196-199: this sentence is confusing, please rewrite or rephrase.
*Rephrased and separate in two sentences, the new paragraph reads as follows:*

"For the present analysis, we define seven domains from north to south; the boundary between domains is defined by a region of roughly 100-200 kilometres that represents the uncertainty in the rupture length of the major events. We consider wider boundaries for the cases of lacking information, in particular in the northern area where the historic record is scarce."

Line 209: Vi to VI

*Corrected*

Line 219: Magnitude Mw 9.3 What reference determines this magnitude? Please incorporate reference or change the magnitude.

*Corrected to 9.5 Mw*

Line 237: remove double parenthesis in "Omori's Law".

*Corrected*

---

## Referee Report (RR1)

[referee-annotated manuscript omitted]

---

## Referee Report (RR2)

Dear Editor

I have already reviewed the article, and I have verified that the authors have made most of the requested corrections, so for me the article is ready to be accepted and continue with the publication process. However, and as a recommendation, the authors to be more careful when talking about surface seismicity associated with these translithospheric structures/faults. If they are going to continue working on this type of structures, please be more rigorous when talking about seismicity and not just naming, without clear evidence of seismicity, that these faults present seismicity.

Best regard and thanks for all.

---

## Author Response (AR2)

September 04 2024

Dear Editor:

Attached I am sending the final version without and with control changes suggested by reviewer 2. Since no figures nor supplementary material has been suggested I only send these two new files and kept the other files from the last version.

In this new version we also addressed the remarks made previously by the editorial support team:

- Format of reference list: formatted accordingly with the guidelines
- Figure 5 panels a, b, c: modified to figures 5, 6, and 7, thus figure 6 and 7 are now figures 8 and 9
- Table contains coloured cells: modified to white background

Best regard,

Gonzalo Yanez